# Assessment of Gliomas’ Grade of Malignancy and Extent of Resection Using Intraoperative Flow Cytometry

**DOI:** 10.3390/cancers15092509

**Published:** 2023-04-27

**Authors:** George Vartholomatos, Georgios S. Markopoulos, Eyrysthenis Vartholomatos, Anna C. Goussia, Lefkothea Dova, Savvas Dimitriadis, Stefania Mantziou, Vaso Zoi, Anastasios Nasios, Chrissa Sioka, Athanasios P. Kyritsis, Spyridon Voulgaris, George A. Alexiou

**Affiliations:** 1Neurosurgical Institute, University of Ioannina School of Medicine, 45110 Ioannina, Greece; gvarthol@gmail.com (G.V.); geomarkop@gmail.com (G.S.M.); eyrys.varth@gmail.com (E.V.); theadova@yahoo.gr (L.D.); savvas_dimitriadis@yahoo.gr (S.D.); stefanimantz@gmail.com (S.M.); vasozoi95@gmail.com (V.Z.); anasiosmd@gmail.com (A.N.); csioka@uoi.gr (C.S.); thkyrits@uoi.gr (A.P.K.); svoulgar@uoi.gr (S.V.); 2Haematology Laboratory, Unit of Molecular Biology and Translational Flow Cytometry, 45110 Ioannina, Greece; 3Department of Pathology, Ioannina University Hospital, 45500 Ioannina, Greece; agoussia@uoi.gr; 4Department of Pathology, German Oncology Center, 4108 Limassol, Cyprus; 5Department of Neurosurgery, University Hospital of Ioannina, 45500 Ioannina, Greece; 6Department of Nuclear Medicine, University Hospital of Ioannina, 45500 Ioannina, Greece

**Keywords:** glioma, grade of malignancy, flow cytometry, intraoperative

## Abstract

**Simple Summary:**

Intraoperative Flow Cytometry (iFC) is a new technique that can help assess the malignancy grade, diagnose tumor type, and evaluate resection margins during solid tumor surgery. This study focuses on the role of iFC in grading gliomas and evaluating resection margins. iFC can analyze tissue samples within 5–6 min and was utilized to evaluate samples from patients with gliomas who underwent surgery over an 8-year period. The study found that high-grade gliomas had a significantly higher tumor index than low-grade gliomas. A cut-off value of 17% in the tumor index was identified as being able to accurately differentiate low- from high-grade gliomas. All low-grade gliomas were diploid, while 22 high-grade gliomas were aneuploid. iFC was also able to verify the presence of malignant tissue in every case when evaluating glioma margins. The study concludes that iFC is a promising intraoperative technique for glioma grading and resection margin assessment.

**Abstract:**

Background: Intraoperative Flow Cytometry (iFC) is a novel technique for the assessment of the grade of malignancy and the diagnosis of tumor type and resection margins during solid tumor surgery. Herein, we set out to analyze the role of iFC in the grading of gliomas and the evaluation of resection margins. Material and Methods: iFC uses a fast cell cycle analysis protocol (Ioannina Protocol) that permits the analysis of tissue samples within 5–6 min. Cell cycle analysis evaluated the G0/G1 phase, S-phase, mitosis, and tumor index (S + mitosis phase fraction) and ploidy status. In the current study, we evaluated tumor samples and samples from the peripheral borders from patients with gliomas who underwent surgery over an 8-year period. Results: Eighty-one patients were included in the study. There were sixty-eight glioblastoma cases, five anaplastic astrocytomas, two anaplastic oligodendrogliomas, one pilocytic astrocytoma, three oligodendrogliomas and two diffuse astrocytomas. High-grade gliomas had a significantly higher tumor index than low grade gliomas (median value 22 vs. 7.5, respectively, *p* = 0.002). Using ROC curve analysis, a cut-off value of 17% in the tumor index could differentiate low- from high-grade gliomas with a 61.4% sensitivity and 100% specificity. All low-grade gliomas were diploid. From the high-grade gliomas, 22 tumors were aneuploid. In glioblastomas, aneuploid tumors had a significantly higher tumor index (*p* = 0.0018). Twenty-three samples from glioma margins were evaluated. iFC verified the presence of malignant tissue in every case, using histology as the gold standard. Conclusion: iFC constitutes a promising intraoperative technique for glioma grading and resection margin assessment. Comparative studies with additional intraoperative adjuncts are necessary.

## 1. Introduction

A complex and diverse category of cancers that affect the brain and its supporting systems are known as brain and central nervous system (CNS) cancers. According to the Global Cancer Observatory, in 2020 there were 308,102 new instances of brain and CNS malignancies that had been diagnosed and 251,329 cancer-related deaths globally [1]. Gliomas constitute the most frequent malignant central nervous system (CNS) tumor in adults. Glioblastoma is the most aggressive form (14.3% of all tumors and 49.1% of malignant tumors) and is more frequent in men [2]. The average life expectancy of patients with glioblastoma is 14 to 16 months, in spite of surgical excision, radiotherapy, and chemotherapy. The fifth edition of the World Health Organization (WHO) Classification of CNS Tumors in 2021 incorporated changes in diffuse glioma classification in adults that were driven by *IDH* mutation status [3].

The important factor in the intraoperative management of gliomas is maximizing tumor resection while preserving function and minimizing complications [4]. Some of the intraoperative technologies that can help achieve this goal are techniques for intraoperative diagnosis. Such techniques include, among others, intraoperative MRI, ultrasonography, fluorescence-guided surgery and, lately, intraoperative flow cytometry (iFC) [5,6,7,8,9,10,11].

Intraoperative Flow Cytometry has been introduced as a novel technique mainly for the evaluation of the grade of malignancy and the presence of cancerous tissue in resection margins during solid tumor surgery [12,13]. Among other things, iFC has been evaluated and verified as a valuable tool for the surgeon during the excision of breast cancer, head and neck tumors, gynecological malignancies, bladder cancer, and liver, pancreatic, gastric, and colorectal cancer [14,15,16,17,18,19,20,21,22]. In CNS tumors, iFC permits the differentiation of low- from high-grade tumors in both children and adults, the analysis of neoplastic tissue during stereotactic biopsies, the diagnosis of central nervous system lymphoma, and the assessment of the meningioma grade of malignancy [13,23,24,25,26,27]. 

Grade and margin evaluation are critical aspects of the diagnosis and management of glioma. Gliomas can vary widely in their aggressiveness and response to treatment, and the grade and margin of the tumor can provide valuable information about its behavior [4]. Herewith, we set out to investigate the value of iFC for the assessment of glioma grade and resection margins. Our results support the value of iFC in both malignancy grade prediction and margin status.

## 2. Materials and Methods

### 2.1. Analyzed Population

Patients hospitalized in the Neurosurgical Department of our institution over an 8-year period who underwent surgery for a brain tumor, were suspected for glioma on MRI, and who had a tumor sample available during surgery for intraoperative flow cytometry analysis, were included in the study. Samples from resection margins, when available, were also collected. The investigators that carried out the DNA content analysis were blinded to the preoperative MRI, intraoperative observations, and frozen section analysis data. 

### 2.2. Intraoperative Flow Cytometry and Pathology Analysis

The ‘‘Ioannina Protocol’’, which lasts five minutes from sample receipt and has previously been described in detail, was used to carry out intraoperative flow cytometry analysis [13]. Briefly, a tumor sample with size of 2–5 mm^3^ from the tumor core and samples from resection margins were obtained. Next, the samples were minced with a Medimachine System (BD Bioscience, Franklin Lakes, NJ, USA). The CellTrics filters (Sysmex Flow Cytometry Europe, Norderstedt, Germany) were used to filter the samples and provide a single cell suspension. Cellular suspensions were stained with 125 μg/mL Propidium Iodide (PI) solution. PI is known to bind to cellular DNA and to emit a maximum fluorescence at 617 nm when excited with a standard blue (488 nm) laser. In addition, a control sample was obtained from normal peripheral blood mononuclear cells (PBMCs) that were isolated from whole blood using a Ficoll gradient and stained under the same conditions as the tumor samples. Following PI fluorescence quantification in a FACSCaliburflow cytometer, CellQuest software was used to analyze the G0/G1 phase, S-phase, and mitosis cell fractions, based on manual gating as well as on calculating the tumor index (S + mitosis phase fraction) and ploidy status (DNA index). According to flow cytometry analysis, the tumors were categorized as low-grade (WHO grade I/II) or high-grade (WHO grade III/IV). Tumors that had been diagnosed were categorized based on the World Health Organization (WHO) 2007, 2016 categorization system. Expression of mutant IDH1, the most common IDH gene mutation, was examined with immunohistochemistry using an IDH1R132H antibody in formalin-fixed, paraffin-embedded glioblastoma samples. Our Institutional Review Board approved the study, which was in accordance with the principles of the Declaration of Helsinki.

### 2.3. Statistical Analysis

The Mann–Whitney U test was used to compare the G0/G1, S-phase, mitotic fraction, and tumor index (S + mitosis fraction) of low-grade vs. high-grade gliomas. Receiver operating characteristic (ROC) analysis was used to assess the threshold value separating low-grade from high-grade gliomas. Continuous data are expressed using the mean and standard deviation. The threshold of statistical significance was defined as a probability value less than 0.05. SPSS V.26 (IBM) and Graphpad Prism V 8.4.2 (Graphpad Software, LLC., Boston, MA, USA) software were used to conduct and display the statistical analyses, respectively.

## 3. Results

Eighty-one patients (52 men, 29 women, mean age years range: 20–78) were included in the study. Among them, there were sixty-eight cases of glioblastoma (WHO grade IV), five cases of anaplastic astrocytomas (WHO grade III), two cases of anaplastic oligodendrogliomas (WHO grade III), one case of pilocytic astrocytoma (WHO grade I), three cases of oligodendrogliomas (WHO grade II) and two cases of diffuse astrocytomas (WHO grade II). The samples were processed with intraoperative flow cytometry analysis and taking the histological assessment as gold standard. Two representative analyses for low-grade and high-grade gliomas are presented in Figure 1 and Figure 2, respectively. Twenty-three samples from the glioma resection margins were evaluated. The utilization of iFC verified the presence of malignant tissue in every case, using histology as the gold standard. 

A case of margin evaluation is presented in Figure 3. As can be seen, iFC successfully characterized tumor cells based on a tumor index of 22% and a DNA index of 1.75. By using both indices, the presence of cancer cells could be assessed in respective margins samples. In our case, the gradual reduction in cells in gated markers M2, M3, and M4 (representing hyperploid cancer cells in G1, S, and G2/M cell cycle phases, respectively) is critical for the evaluation and distinction between the positive margins 1, 2, and 3 and the negative margin 4. 

Following iFC analysis, the DNA index (DNAi) was calculated for all samples for the tumor ploidy assessment. The collective results are presented in Figure 4. All tumors that were aneuploids were also high-grade gliomas. The remaining 53 high-grade gliomas were diploid. All low-grade tumors were diploid. Glioblastomas exhibiting aneuploidy had a significantly higher tumor index than diploid glioblastomas (*p* = 0.0018). 

Next, the percentage of cells in each cell cycle phase was calculated. Figure 4 presents the fraction of cells in the G0/G1 phase as well as the remainder cells in S and G2/M, for which their sum is calculated in the tumor index (Ti). For all tumors, the median G0/G1 phase fraction was 78.3, the median S-phase was 6, the median G2/M phase fraction was 10.5 and the median tumor index value was 21. High-grade gliomas had a significantly higher tumor index than low-grade gliomas (median value 22 vs. 7.5, respectively, *p* = 0.002) (Figure 5). Using ROC curve analysis, a cut-off value of 17% in the tumor index could differentiate low- from high-grade gliomas with a 61.4% sensitivity and 100% specificity. The (ROC) curve analysis based on the tumor index for glioma grading is illustrated in Figure 6. There were 59 diploid tumors and 22 aneuploid tumors. The results are summarized in Table 1.

IDH1 status was available in 47/68 (69.1%) glioblastoma cases. Using immunohistochemistry, the IDH1 mutation (IDH1: c.395G > A p.R132) was found in seven (14.9%) cases. There was higher Ti in tumors without IDH1 mutation compared to IDH1 mutated tumors; however, the difference was not statistically significant (median value 22 vs. 10, respectively, *p* = 0.09).

## 4. Discussion

The present study showed that iFC may differentiate high-grade from low-grade gliomas intraoperatively with 61.4% sensitivity and 100% specificity. More importantly, in resection margins glioma tissue could be identified in all cases using iFC. All aneuploid tumors were high-grade gliomas. Thus, the presence of aneuploidy may prove to be a useful marker to exclude a low-grade tumor.

Flow cytometry is an important adjunct in both clinical and basic research settings. DNA content analysis was one of the first applications of flow cytometry, dating back to the 1970s. DNA content analysis in solid tumors was rarely performed and required substantial time [9]. One of the important findings obtained with this technology was that the presence of aneuploidy correlated with solid tumor prognosis [28]. Intraoperative flow cytometry is the results of the efforts from two research groups from Tokyo, Japan, and Ioannina, Greece, each working independently over the past few years, resulting in techniques for quick DNA content analysis [8,13]. These techniques can be performed within minutes (5 to 19 min), do not require any substance to be administered to the patient, are operator-independent and have minimal sample requirements. Furthermore, flow cytometers are widely available and cell cycle analysis is inexpensive. Now, iFC has been developed as a next-generation technique that allows the analysis of DNA content and the cell cycle of cells obtained during surgery to identify cancer cells and evaluate the extent of tumor removal in various types of cancers, and thus it can help improve the accuracy and safety of surgery [10,29]. A limitation of the current study is that other markers could also be analyzed to provide further evidence that would improve tumor characterization. Some probable future targets that it would make sense to analyze intraoperatively with an updated iFC protocol could include cell-cycle markers such as Ki67 or H3S10p, which could provide a more detailed readout of cell cycle phase distribution. Such an analysis would also determine whether the iFC-derived tumor index, which is based on DNA content, correlates well with other cell-cycle markers, and could further improve the reliability and accuracy of iFC in predicting the malignancy of the tumors. Overall, such follow-up experiments would provide a valuable validation of the iFC technique and could help to improve its accuracy and reliability in clinical settings.

There have been few analyses of gliomas by iFC. Tumor samples that were acquired following the excision of 81 intracranial gliomas were examined using iFC by Shioyama et al. [8]. The total number of samples was 328. There were 52 high-grade gliomas and 29 low-grade gliomas. The term “malignancy index” referred to the proportion of cells with higher-than-average DNA content for all cells, an analog to tumor index and DNA index used in the current study. The malignancy index showed significant variations between grade II, III and IV gliomas. Grade II gliomas had a malignancy index of 13.3 ± 11.0%, grade III gliomas had a malignancy index of 35.0 ± 21.8%, and glioblastomas had a malignancy index of 46.6 ± 23.1%. Additionally, high-grade tumors were more likely to be aneuploid. More specifically, aneuploidy was found in nearly half of grade III tumors, one-third of grade II tumors, and 58.6% of glioblastomas [8]. The findings of the present study are in accordance with Shioyama et al. [8]. We also found that high-grade tumors were more aggressive based on both the flow cytometric metrics tumor index and the DNA index, compared to low-grade gliomas. Additionally, aneuploidy was found in high-grade gliomas. 

In the present study, no low-grade tumors were aneuploid. Nevertheless, Suzuki et al. evaluated 102 consecutive cases of newly diagnosed WHO grade II supratentorial gliomas and found one third of cases to be aneuploid [30]. An important finding was that aneuploidy was more frequent in diffuse astrocytomas than oligodendrogliomas. Diffuse astrocytomas are usually more aggressive than oligodendrogliomas. Diploid tumors showed significantly longer progression-free and overall survival, whereas aneuploid tumor more frequent progressed and dedifferentiated to glioblastoma [30]. The number of low-grade gliomas included in our study may represent a limitation to verify such results. We believe that our ongoing clinical analysis of such cases might offer us the opportunity to verify Suzuki et al.’s findings in the local population.

Saito et al. described an additional role of iFC in characterizing gliomas. In a study of 102 patients with glioblastoma who underwent iFC analysis and received the standard treatment protocol, a correlation with overall survival was performed. Glioblastomas with a high malignant index exhibited better survival only among glioblastoma patients that received radiotherapy plus concomitant adjuvant chemotherapy with temozolomide. Furthermore, the malignant index was correlated with *IDH1* mutation status [31]. A long-term analysis of the clinical status of the presented cases will offer the potential to assess the prognostic role of iFC based on the Ioannina protocol in a future study.

In conclusion, intraoperative flow cytometry is a novel, operator independent and low-cost technique that can be implemented during glioma surgery. One discussed limitation of the present study was that the number of low-grade gliomas included was limited. The inclusion of more low-grade gliomas may further increase the overall sensitivity of this method, and on the other hand may identify some low-grade gliomas that may be aneuploid. Nevertheless, the results showed that iFC may have a role in the discrimination of low- from high-grade glioma and, more importantly, in the evaluation of the peripheral margin’s burden of cancerous tissue. This information is crucial for the surgeon to modify surgical strategy. Further studies are needed to verify our results and correlate flow cytometry metrics with progression-free and overall survival rates of glioma patients.

## 5. Conclusions

The importance of evaluating glioma malignancy by iFC lies in the fact that the accurate determination of tumor grade and malignancy can help to guide treatment decisions and improve patient outcomes. Gliomas can range from low-grade (slow growing) to high-grade (fast growing), and the grade of the tumor can impact the choice of treatment, including surgery, radiation therapy, and chemotherapy. iFC can provide real-time information regarding the tumor’s cellularity, proliferation rate, and other factors that can help to determine its malignancy. This information can help to guide the surgeon in deciding the extent of the tumor resection, and can also help to inform the patient’s postoperative treatment plan. Overall, the use of intraoperative flow cytometry in the evaluation of glioma malignancy can improve the accuracy of tumor grading, the evaluation of margins, and help to guide treatment decisions, ultimately leading to better outcomes for patients with gliomas.

## Figures and Tables

**Figure 1 cancers-15-02509-f001:**
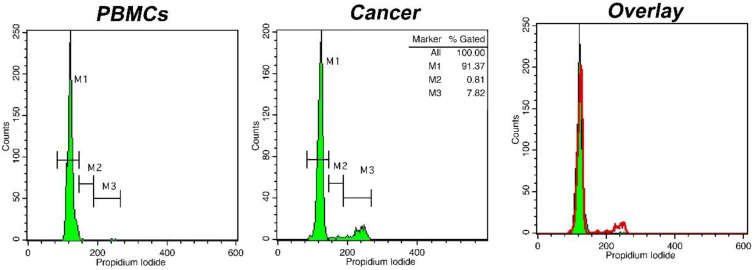
DNA analysis with intraoperative Flow Cytometry in a case of an oligodendroglioma (WHO grade II). Histograms represent DNA content distributions. Markers M1, M2, and M3 correspond to cells in phase G0/G1, S, and G2/M, respectively. Left: distribution of peripheral blood mononuclear cells/PBMCs (solid green in the right overlay), middle: distribution of cancer cells (presented in red in the right overlay). The presented case is diploid, with a DNA index = 1 and a tumor Index of ~8%; right: overlay of two previous histograms.

**Figure 2 cancers-15-02509-f002:**
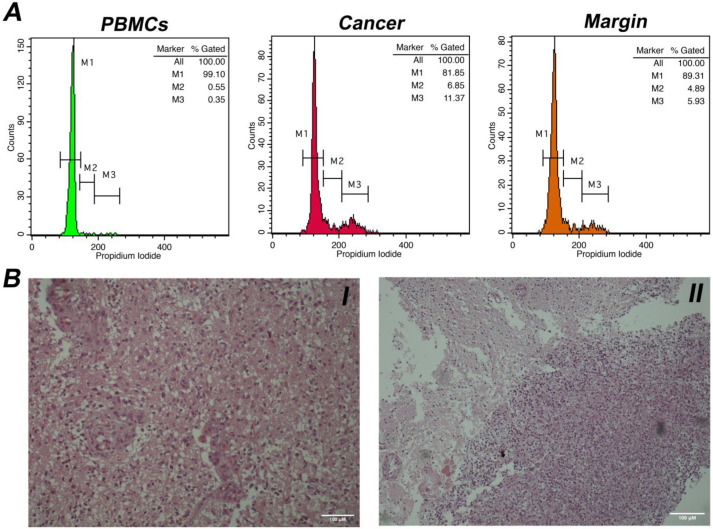
(**A**). DNA analysis with intraoperative Flow Cytometry in a case of a glioblastoma (WHO grade IV). Markers M1, M2, and M3 correspond to cells in phase G0/G1, S, and G2/M, respectively. Left histogram: distribution of peripheral blood mononuclear cells/PBMCs (in green), middle histogram: distribution of malignant cells (in red). The presented case is diploid, with a DNA index = 1 and a tumor index of ~18%; right histogram: a tumor margin (in orange) with index of ~11%. (**B**). I. Histological section of a case of glioblastoma (Hematoxylin & Eosin stain, original magnification × 100). II. Neoplastic cells infiltrate normal glial tissue at the peripheral borders of the tumor (Hematoxylin & Eosin stain, original magnification ×40). Scale bars (100 μM) are present in lower right part of each figure.

**Figure 3 cancers-15-02509-f003:**
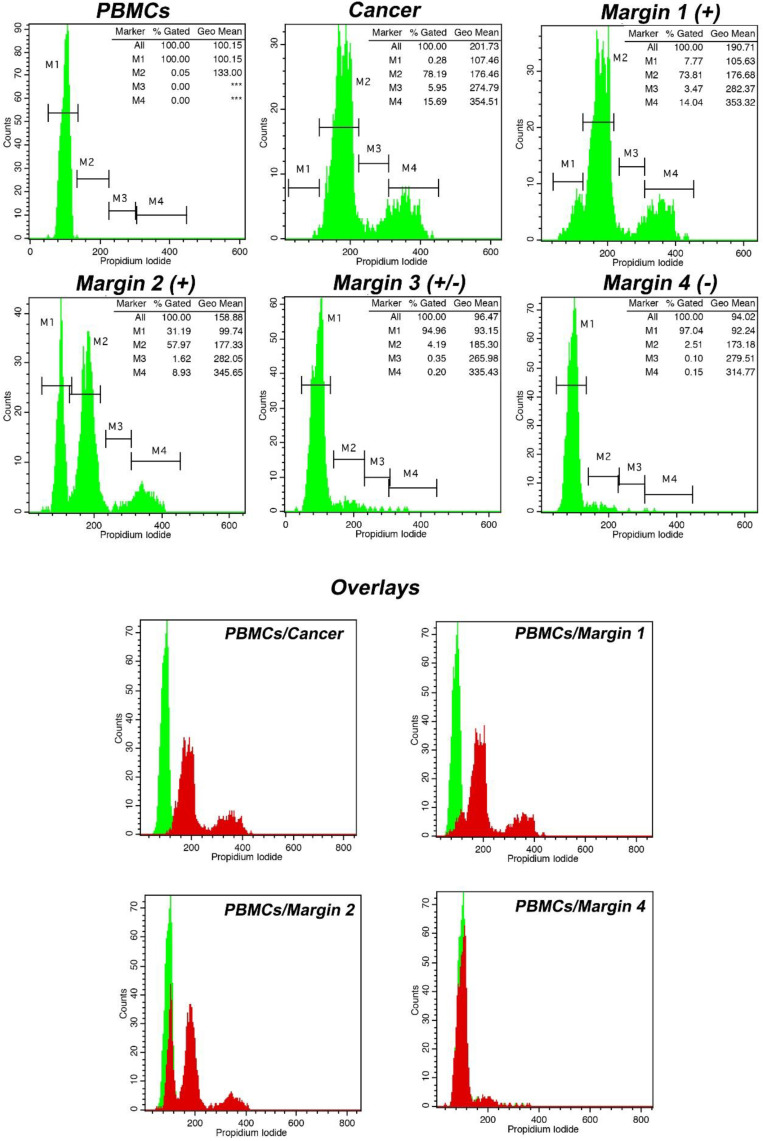
Tumor characterization and margin evaluation in a case of high-grade glioma using intraoperative flow cytometry (iFC). Following PI-staining, DNA was quantified in a flow cytometer. DNA quantity in cell populations of peripheral blood mononuclear cells (PBMCs), cancer, and margins are presented in respective plots. Marker M1 in PBMCs sample denotes cells in G0/G1 cell cycle phase. In cancer sample, Markers M2, M3, and M4 correspond to G0/G1, S, and G2/M cell cycle phases. Tumor index and DNA index quantified, based on the results obtained by iFC and presented in the upper right in each respective fluorescence plot (*** represents absence of cells in the respective marker). In our case, the tumor index has been calculated as ~22% (proportion of cells in S and G2/M). DNA index, a measure of the DNA content and ploidy status, having normal G0/G1 of PBMCs as a reference, was calculated as ~1.75, meaning that the tumor is hyperploid. The gradual reduction in tumor index and/or DNA index in margin samples denotes margin status. The margins are sorted from positive to negative. The lower panel presents overlays of different margin samples regarding normal cells. Margin samples are represented in red in the overlay with green sample representing normal PBMCs.

**Figure 4 cancers-15-02509-f004:**
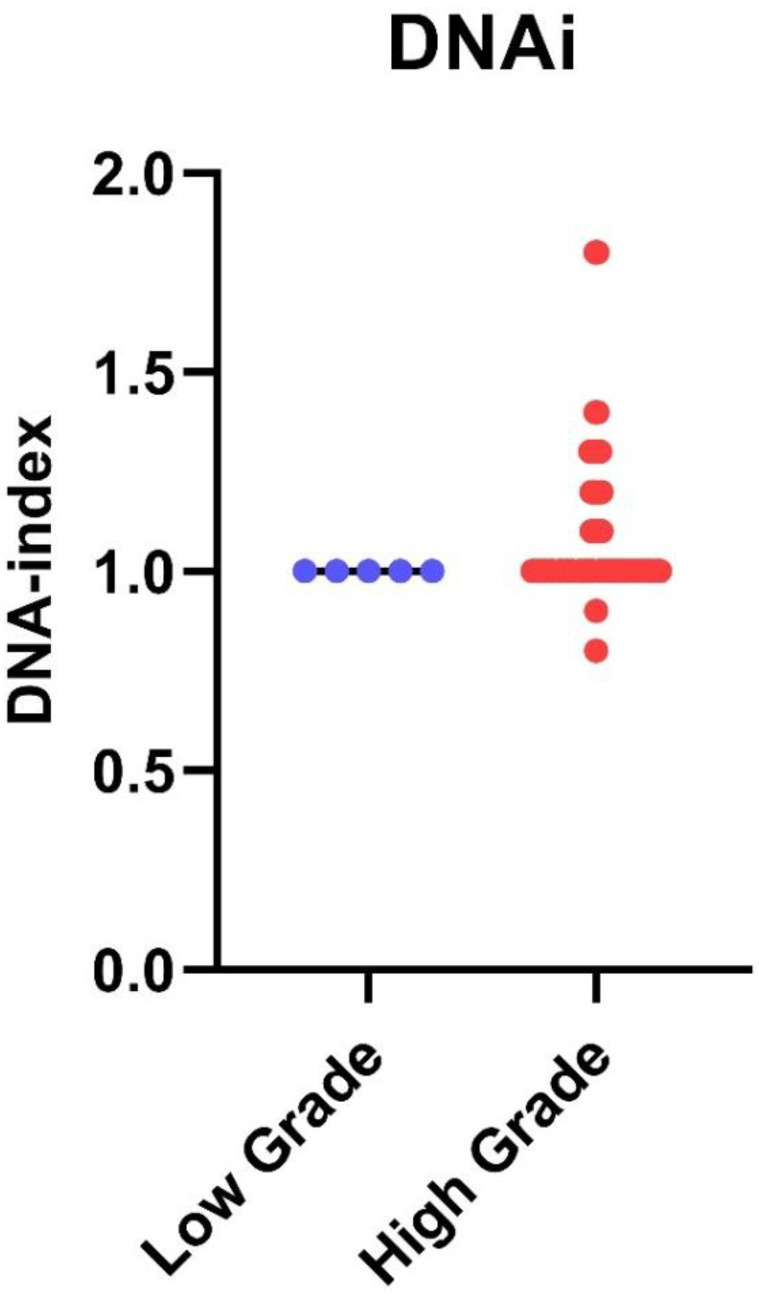
DNA index in low-grade versus high-grade gliomas: DNA index from individual cases has been quantified by iFC as the geometric mean of G0/G1 in cancer cells divided by that of normal diploid cells. DNA index is presented as blue or red dots, for low- and high-grade gliomas, respectively. Median DNA index is shown as a horizontal line in each group. A DNA index of ≠1 has been found only in high-grade gliomas.

**Figure 5 cancers-15-02509-f005:**
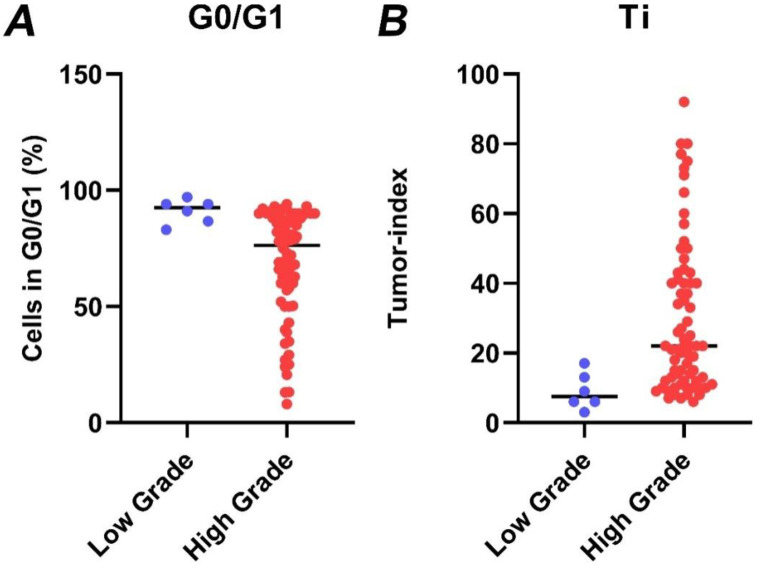
The percentage of cells in G0/G1 (**A**) and the respective tumor index (**B**) in low-grade (blue dots) versus high-grade gliomas (red dots), as quantified by iFC. Tumor index has been quantified in individual samples as the cumulative percentage of cells in S and G2/M cell cycle phases. Median percentages are shown as horizontal lines in each group.

**Figure 6 cancers-15-02509-f006:**
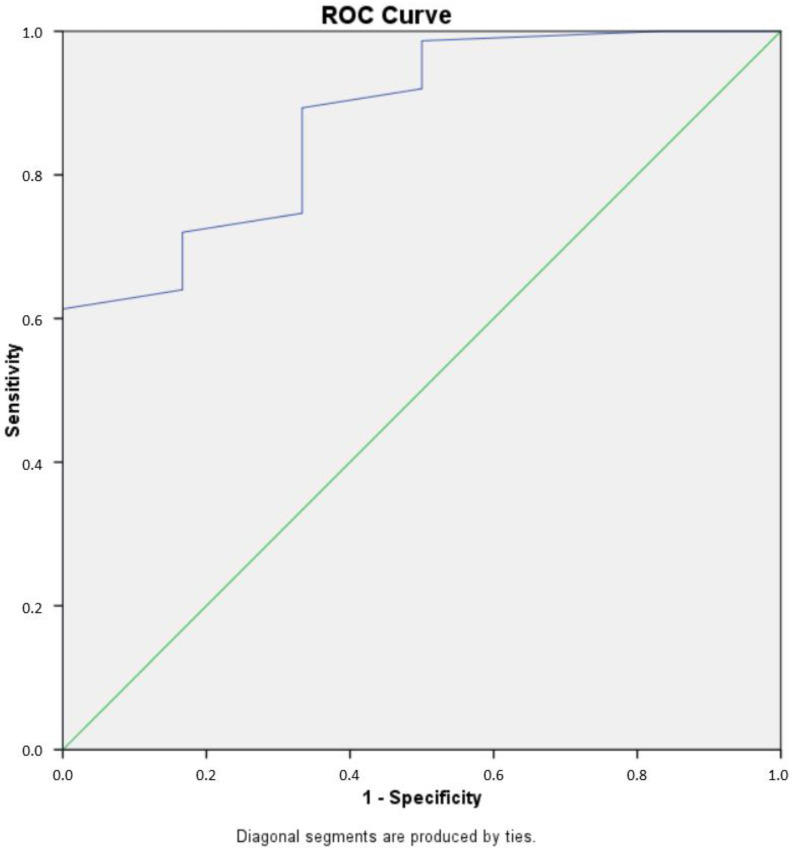
Receiver operating characteristic (ROC) curve analysis of tumor index for glioma grading. The ROC curve illustrates the sensitivity and specificity of iFC-derived tumor index values in differentiating low-grade from high-grade gliomas. The area under the curve (AUC) was 0.876. The optimal cut-off value for tumor index was determined to be 17%. This cut-off value yielded a sensitivity of 61.4% and a specificity of 100% for distinguishing low-grade from high-grade gliomas. The diagonal line on the ROC curve represents a random classifier, while the curve represents the performance of the iFC-derived tumor index in distinguishing between low- and high-grade gliomas. The closer the curve is to the top-left corner, the better the test performs.

**Table 1 cancers-15-02509-t001:** Synopsis of iFC results per analyzed population.

	Low-Grade	High-Grade
No of patients (%)	6 (7%)	75 (93%)
Diploid (%)	6 (10.1%)	53 (89.9%)
Aneuploid (%)	0 (0%)	22 (100%)
G0/G1	91%	67.9%
S-phase	3%	9.4%
G2/M phase	6%	20.7%
Tumor index (S + G2/M)	9%	30.2%

## Data Availability

The data presented are available upon request from the corresponding author.

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
