# Peer review of "Assessment of Gliomas’ Grade of Malignancy and Extent of Resection Using Intraoperative Flow Cytometry"

_cancers, 2023, doi:10.3390/cancers15092509_

Round 1

Reviewer 1 Report

Authors report their experince on the use of intraoperative flow cytone try in a series of brain gliomas. Their experience suggests that iFC is a simple, low cost and almost real time technique to assess radicality and tumor grading during resection. In my opinion the manuscript is clear, concise and well illustrated, I suggest to add data on IDH mutation status to try to correlate this aspect with iFC data.

Author Response

We would like to thank the reviewers for their time and effort to consider our manuscript and for their insightful comments. We revised the manuscript based on their suggestions and now we present a new version for consideration in Cancers Journal.

Please find the responses to reviewers’ comments in Bold, below each respective comment. Changes made to the text are highlighted in yellow.

Reviewer 1

Authors report their experince on the use of intraoperative flow cytone try in a series of brain gliomas. Their experience suggests that iFC is a simple, low cost and almost real time technique to assess radicality and tumor grading during resection. In my opinion the manuscript is clear, concise and well illustrated, I suggest to add data on IDH mutation status to try to correlate this aspect with iFC data.

We would like to express our gratitude to the reviewer for taking the time to evaluate our manuscript and providing us with valuable insights. We have thoroughly considered the reviewer's suggestion and have incorporated the relevant available data on IDH status into the manuscript, which we believe will strengthen the overall quality and significance of our study. Thank you again for your constructive feedback, which has undoubtedly improved the quality of our work.

Reviewer 2 Report

Vartholomatos et al. performed intraoperative flow cytometry on glioma samples during surgery and measured cell-cycle and ploidy indices using DNA content. While the results are generally well presented, iFC has been performed on glioma samples and reported previously in multiple publications. Additionally, the manuscript should be strengthened along these following lines.

1.     The authors used DNA content as a readout of cell cycle phases and ploidy during iFC. While the short time frame of iFC precludes extra staining during operation, the authors should perform control experiments where cell-cycle readout and ploidy is measured using a more accurate readout after the operation. For example, the authors could save some cells during iFC , stain with a cell-cycle marker (e.g. Ki67, H3S10p), and test whether the tumor index measured by >2N DNA correlates well with cell-cycle markers. Or the authors could perform IHC on histological sections measuring Ki67.

2.     The authors noted that using a cutoff of 17% in tumor index could differentiate low from high grade gliomas with a certain sensitivity and specificity. The authors should show the full ROC curve.

3.     Have the authors analyzed cell death using sub-2N DNA content or markers/assays for apoptosis/necrosis? It would be interesting to compare the necrotic core vs other regions of the tumor.

4.     The Simple Summary is not included.

5.     The authors should briefly describe the Ioannina Protocol in the Methods section, including concentration of PI and how 2N/3N/4N DNA was determined from the flow cytometry profile, e.g. were the gates determined manually.

6.     Scale bars should be included in Figure 2B.

7.     Significant proofreading is required, e.g. lines 146~148. Figure 4 is mislabeled as Figure 3 in the legend.

Author Response

We would like to thank the reviewers for their time and effort to consider our manuscript and for their insightful comments. We revised the manuscript based on their suggestions and now we present a new version for consideration in Cancers Journal.

Please find the responses to reviewers’ comments in Bold, below each respective comment. Changes made to the text are highlighted in yellow.

Reviewer 2

Vartholomatos et al. performed intraoperative flow cytometry on glioma samples during surgery and measured cell-cycle and ploidy indices using DNA content. While the results are generally well presented, iFC has been performed on glioma samples and reported previously in multiple publications. Additionally, the manuscript should be strengthened along these following lines.

  1. The authors used DNA content as a readout of cell cycle phases and ploidy during iFC. While the short time frame of iFC precludes extra staining during operation, the authors should perform control experiments where cell-cycle readout and ploidy is measured using a more accurate readout after the operation. For example, the authors could save some cells during iFC , stain with a cell-cycle marker (e.g. Ki67, H3S10p), and test whether the tumor index measured by >2N DNA correlates well with cell-cycle markers. Or the authors could perform IHC on histological sections measuring Ki67.

We express our gratitude to the reviewer for their valuable suggestion. The concept put forward is captivating and has the potential to authenticate our findings in the long run. With the reviewer's recommendation in mind, we are resolute in revising our clinical protocol and patient informed consent to incorporate supplementary stains for cell-cycle markers. Regrettably, carrying out such experiments on the samples that have already been analyzed using the current iFC protocol is presently unfeasible. Nonetheless, we have acknowledged the prospect of additional analyses in the discussion section as a constraint of the current study, and an encouraging avenue for future research.

  1. The authors noted that using a cutoff of 17% in tumor index could differentiate low from high grade gliomas with a certain sensitivity and specificity. The authors should show the full ROC curve.

We agree with the reviewer. We have added the full ROC curve as a new Figure 6.

  1. Have the authors analyzed cell death using sub-2N DNA content or markers/assays for apoptosis/necrosis? It would be interesting to compare the necrotic core vs other regions of the tumor.

We have not analysed cell death and sun-2N DNA and our readout is based  on the percentage of cells in DNA content of cells with intact nucleus. Since the aim of the current study was to correlate DNA content with grade of malignancy and margin status, no markers of apoptosis/necrosis were analysed. However, the suggestion of the reviewer is very interesting for a future study that would evaluate tumor heterogeneity and cell death phenomena and would like to thank the reviewer for the intuitive comment.

  1. The Simple Summary is not included.

Simple summary has been added, based on the reviewer’s suggestion.

  1. The authors should briefly describe the Ioannina Protocol in the Methods section, including concentration of PI and how 2N/3N/4N DNA was determined from the flow cytometry profile, e.g. were the gates determined manually.

We have added the relevant information in the manuscript. Please see the revised text for details.

  1. Scale bars should be included in Figure 2B.

We have added scale bars in Figure 2B, according to the reviewer’s suggestion.

  1. Significant proofreading is required, e.g. lines 146~148. Figure 4 is mislabeled as Figure 3 in the legend.

We would like to thank the reviewer for the suggestion. We have extensively revised the text to correct such mistakes.

Round 2

Reviewer 2 Report

The authors have satisfied my original comments.